# Investigating the Impact of Selective Modulators on the Renin–Angiotensin–Aldosterone System: Unraveling Their Off-Target Perturbations of Transmembrane Ionic Currents

**DOI:** 10.3390/ijms241814007

**Published:** 2023-09-12

**Authors:** Te-Ling Lu, Sheng-Nan Wu

**Affiliations:** 1School of Pharmacy, China Medical University, Taichung 406040, Taiwan; lutl@mail.cmu.edu.tw; 2Department of Research and Education, An Nan Hospital, China Medical University, Tainan 709040, Taiwan; 3School of Medicine, College of Medicine, National Sun Yat-sen University, Kaohsiung 804201, Taiwan

**Keywords:** renin–angiotensin–aldosterone system, small molecules, off-target effect, Na^+^ current, ionic current

## Abstract

The renin–angiotensin–aldosterone system (RAAS) plays a crucial role in maintaining various physiological processes in the body, including blood pressure regulation, electrolyte balance, and overall cardiovascular health. However, any compounds or drugs known to perturb the RAAS might have an additional impact on transmembrane ionic currents. In this retrospective review article, we aimed to present a selection of chemical compounds or medications that have long been recognized as interfering with the RAAS. It is noteworthy that these substances may also exhibit regulatory effects in different types of ionic currents. Apocynin, known to attenuate the angiotensin II-induced activation of epithelial Na^+^ channels, was shown to stimulate peak and late components of voltage-gated Na^+^ current (*I*_Na_). Esaxerenone, an antagonist of the mineralocorticoid receptor, can exert an inhibitory effect on peak and late *I*_Na_ directly. Dexamethasone, a synthetic glucocorticoid, can directly enhance the open probability of large-conductance Ca^2+^-activated K^+^ channels. Sparsentan, a dual-acting antagonist of the angiotensin II receptor and endothelin type A receptors, was found to suppress the amplitude of peak and late *I*_Na_ effectively. However, telmisartan, a blocker of the angiotensin II receptor, was effective in stimulating the peak and late *I*_Na_ along with a slowing of the inactivation time course of the current. However, telmisartan’s presence can also suppress the *erg*-mediated K^+^ current. Moreover, tolvaptan, recognized as an aquaretic agent that can block the vasopressin receptor, was noted to suppress the amplitude of the delayed-rectifier K^+^ current and the M-type K^+^ current directly. The above results indicate that these substances not only have an interference effect on the RAAS but also exert regulatory effects on different types of ionic currents. Therefore, to determine their mechanisms of action, it is necessary to gain a deeper understanding.

## 1. Introduction

The renin–angiotensin–aldosterone system (RAAS) plays a crucial role in regulating various physiological processes in the body, primarily related to blood pressure and fluid balance. It is a complex hormonal system that involves the interaction of several hormones and enzymes [1]. When blood pressure drops or blood flow to the kidneys is reduced, the RAAS is activated to help restore blood pressure and maintain adequate perfusion to vital organs. The dysregulation of this system can lead to various medical conditions [2,3]. For example, overactivation of the RAAS is associated with hypertension, heart failure, and kidney disease [1,3,4,5]. Alternatively, inhibiting certain components of the RAAS, such as ACE inhibitors or blockers of angiotensin II type 1 (AT1) receptors, is a common approach in managing hypertension and heart failure [6].

This review paper aims to showcase a variety of compounds or drugs originally designed to regulate the RAAS. However, in addition to their primary intended actions, these compounds also exhibit regulatory effects on cellular membrane ion channels, specifically through off-target actions. This paper will delve into the diverse range of such compounds, highlighting their interactions with ion channels and the potential implications of these off-target effects. Table 1 displays the chemical structures of the compounds and drugs discussed in this article. Some of these agents are already in clinical use, while others are still in the developmental stage. Some are merely experimental research reagents. Meanwhile, Table 2 presents the abbreviations of these compounds and drugs, as well as their documented off-target effects on ion currents, along with their respective reference citations. The use of these compounds or medications can potentially lead to an overinterpretation of their primary effects as being caused by an influence on the RAAS. Similarly, in the central nervous system, there is believed to also be a RAAS. The effects of these drugs and compounds on the brain may also be influenced by their interactions with ion channels in different cell types.

The following content describes the actions of these compounds and drugs in sequence. In addition to affecting the RAAS, it has been discovered that they also have off-target effects on various ion channels, leading to unintended or unexpected effects.

## 2. Apocynin (Acetovanillone, 4′-Hydroxy-3′-methoxyacetophenone)

Apocynin (aPO), a polyphenolic compound, is a naturally occurring ortho-methoxy-substituted catechol isolated from a variety of plant sources, including *Apocynum cannabinum*, *Pierorhiza kurioa*, and so on [16]. This compound has been used as a selective inhibitor of NADPH-dependent oxidase (NOX) [17]. The excessive or dysregulated production of reactive oxygen species caused by NOX activity can be detrimental to cells and tissues. aPO is recognized as one of the most promising compounds in a variety of pathophysiological disorders, such as inflammatory and neurodegenerative diseases, glioma, and cardiac failure [16,17]. Previous studies have shown the ability of aPO to attenuate the angiotensin II-induced activation of epithelial Na^+^ channels in human umbilical vein endothelial cells and to blunt the activation of these channels caused by epidermal growth factor, insulin growth factor-1, and insulin [18,19].

However, recent studies have shown that this compound can result in the differential stimulation of peak or late-amplitude *I*_Na_ activated by rapid step depolarization in pituitary tumor (GH_3_) cells [7]. aPO-accentuated *I*_Na_ was reversed by ranolazine, an inhibitor of late *I*_Na_. Additionally, the aPO presence could increase the high- or low-threshold amplitude of persistent *I*_Na_ (*I*_Na(P)_) elicited by the isosceles-triangular ramp at either the upsloping or downsloping limb, respectively [7]. These findings thus allow us to see that aPO-stimulated changes in the amplitude, gating, and voltage-dependent hysteresis of *I*_Na(P)_ appear to be unlinked to and upstream of its inhibitory action on NOX activity. These effects, including the attenuation of angiotensin II actions [18,19], could potentially participate in adjustments of varying functional activities in electrically excitable cells (e.g., GH_3_ cells), presuming that similar in vivo findings exist.

From previous pharmacokinetic studies on mice [20], following the intravenous injection of aPO (5 mg/kg), the peak plasma aPO level was detected at 1 min, reaching around 33.1 μM. aPO was reported to be a selective inhibitor of NOX2 activity with an IC_50_ of 10 μM. Moreover, the IC_50_ value required for the aPO-stimulated peak or late *I*_Na_ was 13.2 or 2.8 μM, respectively, while the K_D_ value estimated on the basis of a minimal reaction scheme was 3.4 μM [7]. Therefore, it is important to note that the stimulation of *I*_Na_ by aPO tends to emerge in a manner largely independent of its inhibitory effect on NOX activity. The aPO molecule can exert an interaction at the binding site(s) residing on Na_V_ channels. This study thus highlights an important alternative aspect that has to be taken into account, inasmuch as there is a beneficial or ameliorating effect from aPO in various pathologic disorders, such as inflammatory or neurodegenerative diseases, and heart failure [16,17].

## 3. Esaxerenone (7α-[(2R,4R,5S,7S)-7-(2-Carboxyethyl)-2,3-dihydroxy-5,6-dimethyl-4-(2-oxo-1-pyrrolidinyl)oxytetrahydro-2H-pyran-4-yl]-5β,6β-dihydrospiro[naphthalene-1(2H),2′-pyrrolizine]-3′,5′-dione)

Esaxerenone (ESAX; Minnebro^®^, CS-3150, XL-560) is a new oral, non-steroidal selective blocker of the activity of mineralocorticoid receptors (MRs). By blocking MRs, ESAX reduces the effects of aldosterone, leading to several important physiological effects, such as blood pressure regulation, diuretic effects, and potassium regulation. This drug has seen increasing use in the management of certain medical conditions related to hormone imbalances and kidney function. These disorders include primary aldosteronism (also known as Conn’s disease), refractory hypertension, chronic kidney disease, diabetic nephropathy, and heart failure [21,22,23,24,25,26].

Of interest, a recent study [8] demonstrated that, despite its effectiveness in antagonizing the activity of MRs, the ESAX presence exerted an immediate depressant action on *I*_Na_ in pituitary GH_3_ cells, together with a significant shortening of the inactivation time course of the current. The strength of voltage-dependent hysteresis of *I*_Na(P)_ evoked by the triangular ramp voltage was also depressed by ESAX’s presence. With the continued presence of tefluthrin, the further addition of ESAX effectively counteracted the tefluthrin-induced stimulation of *I*_Na_. Tefluthrin, a type-I pyrethroid insecticide, is viewed as an activator of *I*_Na_ [27,28]. The distinguishable inhibition of peak and late *I*_Na_ by ESAX may thus be caused by one of several ionic mechanisms underlying its marked perturbations on physiological processes in different excitable cells (e.g., GH_3_ cells), assuming that similar observations exist in vivo.

It also needs to be mentioned that the presence of neither dexamethasone, a synthetic glucocorticoid, nor aldosterone had any effects on *I*_Na_ in GH_3_ cells [8]. However, despite the continued exposure to aldosterone, the subsequent application of ESAX resulted in a further decrease in the amplitude of peak *I*_Na_. It, therefore, seems unlikely that the ESAX-mediated inhibition of the amplitude and gating kinetics of *I*_Na_ was largely associated with its blockade of MRs. It is also important to mention that the maximal plasma concentration of ESAX was reported to reach around 3 μM after a 10-day administration with a dose of 100 mg/d [29]. As such, the inhibitory effect of ESAX on peak and late *I*_Na_ could noticeably be of clinical or therapeutic relevance [23,25]. In concert with its antagonistic action on MRs, ESAX may exert an additional ameliorating action on the salt-induced elevation of blood pressure or on kidney injuries [22,26].

## 4. Dexamethasone (9α-Fluoro-11β,17α,21-trihydroxy-16α-methylpregna-1,4-diene-3,20-dione)

The large-conductance Ca^2+^-activated K^+^ (BK_Ca_) channel is responsive to both membrane depolarization and intracellular Ca^2+^ elevation and is activated during the action potentials of neuroendocrine and endocrine cells. When this channel is opened, K^+^ ions flow out of the cell through the pore, generating an ion current (i.e., a Ca^2+^-activated K^+^ current) that hyperpolarizes the cell. This hyperpolarization subsequently reduces cell excitability, limiting its responsiveness to stimuli [30,31,32]. Hyperpolarization, observed in endocrine or neuroendocrine cells, can thus serve as a negative feedback mechanism, effectively halting exocytosis by turning off voltage-gated Ca^2+^ currents [30,31].

Glucocorticoids have been shown to modify the sensitivity of BK_Ca_ channels to protein phosphorylation in freshly isolated pituitary cells [33]. Previous work demonstrated the ability of cortisol to decrease intracellular Ca^2+^ and prolactin release in pituitary cells [34]. Moreover, another notable study reported that dexamethasone (DEX), a synthetic glucocorticoid, was effective in increasing the open-state probability of BK_Ca_ channels observed in pituitary GH_3_ and AtT-20 cells, which was thought to be through a non-genomic mechanism [9]. DEX was found to increase the open-state probability of BK_Ca_ channels with no change in single-channel conductance. This compound also depressed the firing of spontaneous action potentials in GH_3_ cells through the stimulation of whole-cell Ca^2+^-activated K^+^ currents.

The profile of BK_Ca_ channel subunits present in these endocrine cells appears to be similar because β-subunits are lacking, and α-subunits were found to express the STREX-1 exon [35]. While DEX is primarily used for its anti-inflammatory effects, it can also have some mineralocorticoid (aldosterone-like) effects at higher doses or with prolonged use. DEX can bind to mineralocorticoid receptors (MRs) in the same way as aldosterone does, and it exerts similar effects on the kidneys [36,37,38]. However, the immediate stimulation of BK_Ca_ channels by DEX or other glucocorticoids might partly be responsible for its rapid inhibition of hormone release because such an effect facilitates Ca^2+^-induced feedback hyperpolarization and prevents voltage-activated Ca^2+^ entry [9,31,38,39].

The DEX concentration used to stimulate BK_Ca_-channel activity might not align with the physiological levels of endogenous glucocorticoids that exhibit similar potency. However, during severe injuries, the daily production rate of glucocorticoids can significantly increase, up to at least tenfold. Additionally, previous research has indicated that locally produced steroids in the brain and endocrine glands, known as neurosteroids, may play a role in regulating cell excitability [40,41,42]. The DEX suppression test has been widely used to screen for adrenal hyperfunction because it is a potent synthetic glucocorticoid. High-dosage glucocorticoids have been used for immunologically mediated diseases [43]. The concentrations of plasma glucocorticoids obtained after single infusions can range between 16 and 72 μM [43]. Therefore, the stimulatory properties of BK_Ca_ channels induced by glucocorticoids may be clinically and therapeutically relevant. Glucocorticoids can also be interesting tools used to characterize the properties of BK_Ca_ channels.

## 5. Sparsentan (N-(4-((5-(4-(2-Carboxyethyl)-1-piperazinyl)-2-methylphenyl)sulfonyl)-2,5-dimethylphenyl)-4-(2-methylphenyl)-1-piperazinecarboxamide)

Sparsentan (RE-021, PS433540, DARA-a) is a first-in-class, orally active, dual-acting, selective antagonist of the AT1 receptor and the endothelin type A (ET_A_) receptor [44]. This compound has been in development for the treatment of focal segmental glomerulosclerosis, IgA nephropathy, and other kidney-related conditions [45,46,47]. In focal segmental glomerulosclerosis, certain segments of some glomeruli become scarred and damaged, leading to a decrease in their filtering function. It has been recently shown that both endothelin and angiotensin II can injure podocytes inside the glomeruli and that angiotensin II can exert a physiological function, including an ion-channel-modifying effect [10]. It is, therefore, expected to provide meaningful clinical benefits in mitigating proteinuria, as well as in improving the glomerular filtration rate, although not specifically related to cardiovascular therapy [45,48].

Recent work unveiled that the amplitude of *I*_Na_ recorded from pituitary GH_3_ cells noticeably decreases during cell exposure to sparsentan [10]. The inactivating or deactivating time course of *I*_Na_ evoked by a short depolarizing pulse became concurrently hastened during cell exposure to sparsentan, although the initial rising phase of *I*_Na_ (i.e., the activation time course) remained little influenced. Additionally, on the basis of the estimated inactivation time constants of the current taken during exposure to different sparsentan concentrations, the value of the dissociation constant (K_D_) observed in GH_3_ cells was derived and then yielded at 2.09 μM. This value was noted to be similar to the IC_50_ value (1.21 μM) required for the sparsentan-mediated inhibition of late *I*_Na_, but smaller than what is needed (15.04 μM) for its suppression of peak *I*_Na_ [10]. Furthermore, it was observed that sparsentan, within a concentration range of 0.3–3 μM, had minimal impact on the peak component of *I*_Na_ during depolarizing pulses. However, it effectively blocked late *I*_Na_ measured at the end of these pulses, suggesting the preferential blocking of late *I*_Na_ by sparsentan. During cell exposure to sparsentan, the steady-state inactivation curve of peak *I*_Na_ shifted in a more negative direction along the voltage axis with no apparent changes in the gating charge of the curve. The resurgent or window *I*_Na_, elicited by various ramp-pulse waveforms in GH_3_ cells, was also found to be attenuated upon the addition of sparsentan [10]. However, the mRNA transcripts for the α-subunit of Na_V_1.1, Na_V_1.2, and Na_V_1.6 were previously reported to be expressed in GH_3_ cells [49]. Whether sparsentan’s effectiveness in altering the magnitude and gating kinetics of either *I*_Na_ residing in heart cells (e.g., Na_V_1.5) or other isoforms of Na_V_ channel occurs is still worthy of further exploration.

The existence of sparsentan was also found to directly inhibit the amplitude of delayed-rectifier K^+^ current (*I*_K(DR)_) and *erg*-mediated K^+^ current (*I*_K(erg)_) in GH_3_ cells [10]. Therefore, the combined effectiveness of sparsentan in the antagonism of the endothelin type A receptor and AT1 receptors, as well as the direct inhibition of *I*_Na_, *I*_K(DR)_, and *I*_K(erg)_, may synergistically interfere with the functional activities of electrically excitable cells (e.g., GH_3_ cells) occurring in vivo. Sparsentan may exert repurposing effects on membrane ionic currents, and these actions could be of therapeutic or clinical relevance. However, whether sparsentan or other structurally similar compounds, such as atrasentan and BMS-248367, exert any overarching effects on the magnitude and/or gating of *I*_Na_ or other types of ionic currents warrants further investigation.

## 6. Telmisartan (4′-[[4-Methyl-6-(1-methylbenzimidazol-2-yl)-2-propylbenzimidazol-1-yl]methyl]biphenyl-2-carboxylic acid)

Telmisartan (TEL, Kinzal^®^, Telma^®^, Micardis^®^), a non-peptide, orally active blocker of the angiotensin II type 1 (AT1) receptor, is the newest available drug class for the treatment of hypertension and various types of cardiovascular disorders [50]. TEL is believed to act by selectively blocking AT1 receptors. By doing so, it inhibits the binding of angiotensin II to these receptors, resulting in an impact on the RAAS of the body. This drug has also been demonstrated to exert anti-inflammatory actions, which are closely linked to its activation of peroxisome proliferator-activated receptor (PPAR)-γ activity [51,52]. However, growing evidence indicates that TEL may interact with ion channels to perturb the magnitude and gating kinetics of whole-cell ionic currents. Losartan, another AT1 receptor antagonist, has been shown to modify cardiac-delayed rectifier K^+^ current [53]. Recent findings have demonstrated that TEL produces a stimulatory action on *I*_Na_ in a concentration- and state-dependent fashion in HL-1 atrial cardiomyocytes [11,12,13,14]. The addition of this drug was found to preferentially stimulate late over peak *I*_Na_ in a concentration range from 0.03 to 30 μM (IC_50_ = 1.2 versus 2.1 μM). The results indicate a 10-fold selectivity for its activation of late versus peak *I*_Na_ in HL-1 cells [13].

Plasma TEL concentrations after oral and intravenous administrations of a single dose of 40 mg TEL were previously reported to reach about 0.087 μM and 2.32 μM, respectively [54]. Therefore, in addition to blocking the AT1 receptor, TEL may also directly stimulate *I*_Na_ within the therapeutic range. It is promising that several studies have reported that the existence of intracellular angiotensin II is involved in the effect of TEL on ionic currents [6]. However, with continued exposure to TEL, the addition of angiotensin II (200 nM) was unable to reverse the TEL-mediated stimulation of *I*_Na_ or the suppression of *I*_K(erg)_ residing in HL-1 atrial cardiomyocytes [13]. Further studies are thus required to determine the extent to which TEL, a medication that blocks intracellular AT1 receptors, regulates membranous ionic channels.

The RAAS primarily focuses on its role in the kidneys and the cardiovascular system. However, in recent years. research has revealed that the components of the RAAS are also present and active in certain regions of the brain, leading to the recognition of a brain-specific RAAS. This brain RAAS is involved in various physiological processes, including the regulation of blood pressure, fluid balance, and electrolyte homeostasis, as well as the modulation of neuronal activity, cognition, and behavior [55]. It is important to note that the dysregulation of the brain RAAS has been implicated in several neurological disorders, including hypertension, Alzheimer’s disease, and strokes.

However, another report by Lai et al. demonstrated the effectiveness of TEL in stimulating *I*_Na_ elicited by abrupt membrane depolarization in mHippoE-14 neurons [14]. It has been noted that the TEL-perturbed stimulation of *I*_Na_ is not instantaneous but rather tends to develop over time based on the openness of Na_V_ channels, thereby producing a resultant reduction in current inactivation [11,14]. TEL’s presence enhances *I*_Na_ magnitude with an EC_50_ value of 0.94 μM, suggesting that this drug, when used for the stimulation of peak *I*_Na_, is more potent and efficacious than either tefluthrin or epicatechin-3-gallate [27,56].

Furthermore, Na_V_1.7 was found to be a subfamily of Na_V_ channels functionally expressed in hippocampal neurons [57]. The researchers also proposed that the unique effects of TEL on *I*_Na_ in mHippoE-14 neurons are not associated with a mechanism linked to either binding to AT1 receptors or the activation of PPAR-γ. Valsartan, another blocker of AT1 receptors, also stimulates *I*_Na_ effectively in NSC-34 neuronal cells [12]. On the basis of the two different theoretical models used for investigating TEL effects on simulated *I*_Na_, it is postulated that the TEL molecule acts predominantly on the fast-inactivated states of Na_V_ channels [13].

Because of the importance of Na_V_ channels in contributing to the excitability and automaticity in different types of excitable cells, including heart, neuroendocrine, and endocrine cells and hippocampal neurons, their stimulatory actions on *I*_Na_ may provide notable insights into the pharmacomechanism of TEL effects, both in basic research and clinical practice [13,14,58,59]. An intriguing study demonstrated that TEL has potential benefits for managing Alzheimer’s disease in African Americans [60]. Therefore, additional research is necessary to elucidate the degree to which the effect of TEL or other structurally similar drugs on cognitive function can be attributed to their interactions with ion channels, especially Na_V_ channels.

## 7. Tolvaptan (N-[4-(7-Chloro-2,3,4,5-tetrahydro-5-hydroxy-1H-1-benzazepin-1-yl)-3-methylbenzoyl]benzene sulfonamide)

Tolvaptan (TLV, Samsca^®^ or Jinarc^®^) is recognized as an oral aquaretic agent that functions as a selective, competitive antagonist of the vasopressin V_2_ receptor, used to treat different types of hyponatremia associated with varying pathological conditions, such as congestive heart failure, cirrhosis, or the syndrome of inappropriate antidiuretic hormone [61,62,63,64]. The binding of vasopressin (also known as antidiuretic hormone or ADH) to the vasopressin V_2_ receptor plays a crucial role in regulating water reabsorption in the kidneys. When vasopressin binds to V_2_ receptors, it triggers a signaling cascade inside the cells of renal collecting ducts in the kidneys. The syndrome of inappropriate antidiuretic hormone secretion is a medical condition in which the body releases too much antidiuretic hormone. The excess of antidiuretic hormone leads to increased water retention in the kidneys and the dilution of electrolytes in the bloodstream. The presence of TLV, as a vasopressin receptor antagonist, can interfere indirectly with the RAAS by increasing urine output and triggering a compensatory response involving the release of renin, angiotensin II, and aldosterone [1,64]. It was previously noted to be effective at improving the hyponatremic conditions that may occur in different pathologic conditions, such as those following pituitary surgery [65]. It also needs to be mentioned that, in Madin–Darby canine kidney (MDCK) cells, TLV exposure was effective in suppressing *I*_K(DR)_ elicited by a ramp pulse [15].

Tricyclic antidepressants such as imipramine have been demonstrated to suppress various types of K^+^ currents, including *I*_K(DR)_ and *I*_K(M)_ [66,67]. A recent study also demonstrated that TLV effectively and differentially suppressed the amplitude of *I*_K(DR)_ and *I*_K(M)_ in a concentration- and time-dependent fashion in pituitary GH_3_ cells [15]. The IC_50_ values of TLV required to suppress *I*_K(DR)_ and *I*_K(M)_ in these cells were estimated as 6.42 and 1.91 μM, respectively. These results demonstrate that TLV exhibits a preferential inhibitory effect on *I*_K(M)_ compared with *I*_K(DR)_. However, in GH_3_ cells preincubated with vasopressin (1 μM), the inhibitory effect of TLV on these K^+^ currents remained unaffected. Therefore, whether an alternative effect on the antidepressant and anxiolytic actions exerted by the blockers of the vasopressin V_1B_ receptor, as described previously [68], is linked to their possible actions on multiple types of K^+^ currents remains to be essentially resolved.

Therefore, this additional aspect should be noticeably considered when assessing the aquaretic effect of TLV or its structurally similar compound [62]. It is thus tempting to speculate that both the direct inhibition of multiple K^+^ currents and blockades of the vasopressin V_2_ receptor by TLV or other structurally similar non-peptide compounds produce beneficial effects on patients with hyponatremia or the syndrome of inappropriate antidiuretic hormone secretion [64,65].

## 8. Conclusions

When assessing the impact of the aforementioned compounds and drugs on the RAAS [5], it is important to be cautious about the potential over- or under-interpretation of outcomes. This caution is necessary because these substances may have off-target effects on ionic currents (Figure 1), potentially influencing the results and leading to misinterpretation. Moreover, for example, in patients who are already taking ranolazine to treat chronic angina pectoris, if they simultaneously take esaxerenone (ESAX) or sparsentan, there will be a synergistic inhibitory effect on late *I*_Na_ in different excitable cells. This is because ranolazine is considered a late *I*_Na_ blocker [69,70]. Perhaps by utilizing different (whole-animal or cell-specific) knockout or knockdown models related to the RAAS [71,72] and simultaneously conducting in-depth comparisons with the use of these abovementioned substances, it will be possible to identify more detailed mechanisms of action for these substances. Many experimental results shown here were derived from pituitary tumor (GH_3_) cells. Hence, whether similar findings will also occur in other types of excitable cells remains to be seen.

The Na^+^ load itself does indeed impact the activity of the RAAS and its regulatory effects. Therefore, investigating how the sodium load affects the actions of the drugs and compounds presented herein in patients with chronic kidney disease is worthy of further research. Additionally, the RAAS and circadian rhythms, or other biorhythms of other hormones, may also be important topics. It is also worth further investigating whether the drugs or compounds presented in this paper will affect these rhythms.

## Figures and Tables

**Figure 1 ijms-24-14007-f001:**
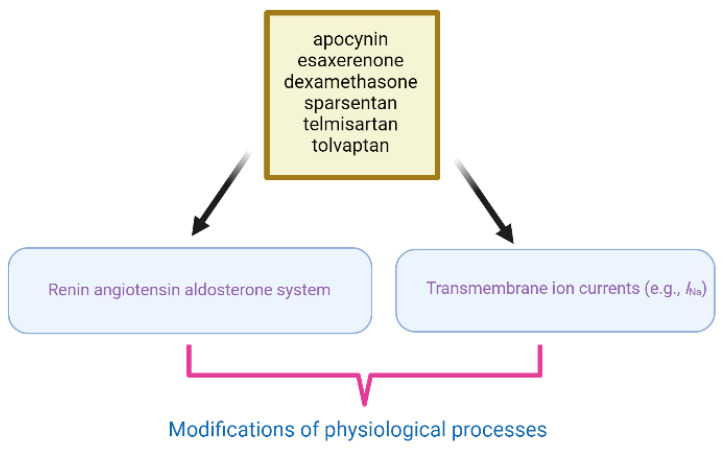
Illustration of the effects of apocynin, esaxerenone, dexamethasone, sparsentan, telmisartan, and tolvaptan on the renin–angiotensin–aldosterone system (RAAS). These compounds also exert regulatory effects on cell-membrane ion currents (e.g., sodium current (*I*_Na_)), resulting in cumulative impacts on numerous physiological processes.

**Table 1 ijms-24-14007-t001:** Chemical structures of the compounds and drugs described in this paper. The chemical structures of these compounds and drugs are taken from the PubChem website (https://pubchem.ncbi.nlm.nih.gov/, accessed on 1 August 2023).

Drug or Compound	Chemical Structure
Apocynin	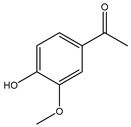
Esaxerenone	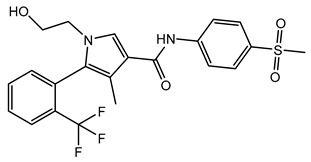
Dexamethasone	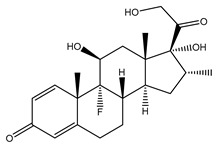
Sparsentan	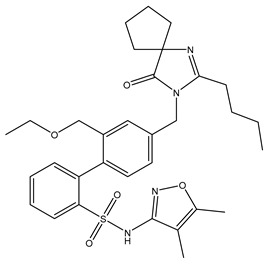
Telmisartan	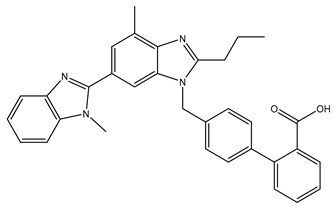
Tolvaptan	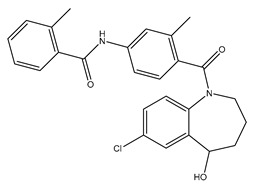

**Table 2 ijms-24-14007-t002:** Off-target effects of the compounds and drugs on transmembrane ionic currents.

Drug or Compound	Abbreviation	Off-Target Effect on Ionic Currents	References
Apocynin	aPO	↑*I*_Na_	[7]
Esaxerenone	ESAX	↓*I*_Na_	[8]
Dexamethasone	DEX	↑*I*_K(Ca)_	[9]
Sparsentan	RE-021 *	↓*I*_Na_, ↓*I*_K(erg)_, ↓*I*_K(DR)_	[10]
Telmisartan	TEL	↑*I*_Na_, ↓*I*_K(erg)_	[11,12,13,14]
Tolvaptan	TLV	↓*I*_K(DR)_, ↓*I*_K(M)_	[15]

↑ refers to an increase in current amplitude, while ↓ indicates a decrease in the amplitude. * This article directly describes the use of sparsentan and does not involve the use of RE-021.

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
