# Peer review of "Investigating the Impact of Selective Modulators on the Renin–Angiotensin–Aldosterone System: Unraveling Their Off-Target Perturbations of Transmembrane Ionic Currents"

_ijms, 2023, doi:10.3390/ijms241814007_

Round 1
Reviewer 1 Report
Very interesting topic,
However I do miss some points that should be taken in the discussion, when we talk about RAAS and RAAS inhibition
- The PK/PD characteristics of the mentionned drugs, who are often not covering 24H
-The sodium load, what gives major influence in RAAS activity and RAAS blockade, both in intake, as in CKD what is the case in CKD
- circadian rythms of RAAS and other hormones, and several biorhythms, where it is now really described as if they do not exist
Author Response
Thanks for the reviewer’s comment.
However I do miss some points that should be taken in the discussion, when we talk about RAAS and RAAS inhibition
- The PK/PD characteristics of the mentionned drugs, who are often not covering 24H
Ans: Thanks for the reviewer’s comments.
Pharmacokinetcs (TK) refers to the study of how a drug moves through the body. This includes process such as absorption, distribution, metabolism, and excretion. Pharmacodynamics (PD) deals with the study of how a drug interacts with its target receptors or molecules in the body and how these interactions lead to therapeutic of adverse effect. Together, PK and PD are essential in the field of pharmacology to optimize the efficacy, and safety of drugs.
Pharmacokinetics and pharmacodynamics are indeed very complex, and they are not the main topics extensively discussed in this paper. They will have to be left for further in-depth exploration in the future.
-The sodium load, what gives major influence in RAAS activity and RAAS blockade, both in intake, as in CKD what is the case in CKD
Ans: Thanks for the insightful comments pointed out by the reviewer. We decide to included this sentence into the revised manuscript. That is, “The Na load itself does indeed impact the activity of the RAAS and its regulatory effects. Therefore, investing how sodium load affects the actions of these drugs or compounds presented herein in patients with chronic kidney disease is worthy of further research.” (indicated in lines 343-346 in the revised manuscript.
- circadian rythms of RAAS and other hormones, and several biorhythms, where it is now really described as if they do not exist
Ans: Thanks for the comment provided by the reviewer. An another statement regarding this issue was hence included the revised manuscript (indicated in lines 346-348).

Reviewer 2 Report
This is a very interesting, detailed and well-constructed review on the literature describing the effects of various mainstream drugs on ion currents. Anecdotally it is known that people do better on certain drugs versus the others and one of the underlying issues may be exactly what the authors describe here. The support for the effect of these drugs on various ion currents is provided. However, most of these studies involve in vitro work, often in transformed cells (i.e. cancer cells) and therefore the physiological effects in vivo may be quite different. Please see below my comments:
1. It appears that most of this work was done in the pituitary GH3 cells. Could the authors please comment or discuss the impact this has on the generalizability of the effects?
2. Line 157: could you please specify whether these are primary cells or transformed cells?
3. The first sentence of the conclusion is unclear. Could you perhaps clarify?
Author Response
Reviewer 2
Comments and Suggestions for Authors
This is a very interesting, detailed and well-constructed review on the literature describing the effects of various mainstream drugs on ion currents. Anecdotally it is known that people do better on certain drugs versus the others and one of the underlying issues may be exactly what the authors describe here. The support for the effect of these drugs on various ion currents is provided. However, most of these studies involve in vitro work, often in transformed cells (i.e. cancer cells) and therefore the physiological effects in vivo may be quite different. Please see below my comments:
Ans: Thanks for the valuable comments provided by the reviewer.
- It appears that most of this work was done in the pituitary GH3 cells. Could the authors please comment or discuss the impact this has on the generalizability of the effects?
Ans: Thanks for the insightful comments pointed out by the reviewer. We still need to clarify some matters: many experimental results were derived from pituitary tumor (GH3) cells. Whether similar results will also occur in other types of excitable cells hence remains to be further researched. This description was also included in the revised manuscript for the perusal (indicated in lines 339-342).
- Line 157: could you please specify whether these are primary cells or transformed cells?
Ans: Thanks for the reviewer’s comment. The sentence was hence appropriately changed to “Glucocorticoids have been shown to modify the sensitivity of BKCa channels to protein phosphorylation in freshly isolated pituitary cells”. (indicated in lines 156-157 of the revised manuscript).
- The first sentence of the conclusion is unclear. Could you perhaps clarify?
Ans: As advised by the reviewer, the sentences were hence appropriately changed to “When assessing the impact of the compounds or drugs mentioned on the RAAS [5], it is important to be cautious about potential over- or under-interpretation of outcomes. This caution is necessary because these substances may possess off-target effects on ionic currents, potentially influencing the results and leading to misinterpretation.” (indicated in lines 329-332 of the revised manuscript).
